# Localization and Bioreactivity of Cysteine-Rich Secretions in the Marine Gastropod *Nucella lapillus*

**DOI:** 10.3390/md19050276

**Published:** 2021-05-15

**Authors:** Mariaelena D’Ambrosio, Cátia Gonçalves, Mariana Calmão, Maria Rodrigues, Pedro M. Costa

**Affiliations:** UCIBIO–Applied Molecular Biosciences Unit, Departamento de Ciências da Vida, Faculdade de Ciências e Tecnologia da Universidade Nova de Lisboa, 2829-516 Caparica, Portugal; cv.goncalves@campus.fct.unl.pt (C.G.); m.calmao@campus.fct.unl.pt (M.C.); mv.rodrigues@campus.fct.unl.pt (M.R.)

**Keywords:** marine biotechnology, bioreactives, toxicity, thiols, dogwhelk, Gastropoda

## Abstract

Marine biodiversity has been yielding promising novel bioproducts from venomous animals. Despite the auspices of conotoxins, which originated the paradigmatic painkiller Prialt, the biotechnological potential of gastropod venoms remains to be explored. Marine bioprospecting is expanding towards temperate species like the dogwhelk *Nucella lapillus*, which is suspected to secrete immobilizing agents through its salivary glands with a relaxing effect on the musculature of its preferential prey, *Mytilus* sp. This work focused on detecting, localizing, and testing the bioreactivity of cysteine-rich proteins and peptides, whose presence is a signature of animal venoms and poisons. The highest content of thiols was found in crude protein extracts from the digestive gland, which is associated with digestion, followed by the peribuccal mass, where the salivary glands are located. Conversely, the foot and siphon (which the gastropod uses for feeding) are not the main organs involved in toxin secretion. Ex vivo bioassays with *Mytilus* gill tissue disclosed the differential bioreactivity of crude protein extracts. Secretions from the digestive gland and peribuccal mass caused the most significant molecular damage, with evidence for the induction of apoptosis. These early findings indicate that salivary glands are a promising target for the extraction and characterization of bioactive cysteine-rich proteinaceous toxins from the species.

## 1. Introduction

In the animal kingdom, toxin secretion and delivery systems are key evolutionary traits [1,2]. Venoms and poisons are constituted by complex mixtures of bioactives, from toxins to permeabilizing enzymes and co-factors, salts, and small peptides [3,4] that have been recruited into poisonous secretions through intricate evolutionary processes [2]. These bioreactive compounds are mostly used for defense, competition, and predation [5], and, due to their potential specificity towards a wide range of molecular targets, they gave rise to a considerable interest for drug discovery purposes [6]. Venomous animals like snakes, scorpions, and spiders have been receiving attention for their potential biomedical applications and were recently joined by marine invertebrates like cnidarians, annelids, cone snails, and even cephalopods [7,8,9].

Cysteine-rich secretory proteins, also known as CRISPs, are low molecular mass proteins commonly found in animal venoms, being well described in snakes [10,11]; they are characterized by multiple thiol-bearing motifs, which render them highly reactive due to their high nucleophilicity [12]. Cysteine-rich secretions are found in many organisms, marine included, even though their functions are not entirely understood [10]. They have been found in the buccal glands of the hematophagous lamprey *Lethenteron japonicum* (an ectoparasite of fish), for instance, and act as vasodilators by blocking calcium channels [13]. As another example, novel studies have shown the presence of cysteine-rich neurotoxins secreted by the proboscis’ predatorial Polychaeta *Eulalia* sp. [14]. These induce neuromuscular paralysis in their prey (typically mollusks, barnacles, and other Polychaeta) [15]. Still, a group of cysteine-rich toxins that has been relatively well explored for biotechnological purposes is that of conotoxins, which are small proteinaceous highly hazardous toxins secreted by diverse species of the predatory snails *Conus*, many of which are lethal to humans [16]. Most conotoxins are characterized for their interaction with voltage-gated Na^+^, K^+^, and Ca^2+^ channels [17,18]. Ziconotide, commercialized as Prialt, is an approved drug developed from the ω-conotoxin MVIIA of the fish-hunting cone snail *Conus magnus*, which is a potent non-opioid analgesic with a stronger effect than morphine [19]. This protein selectively and reversibly blocks *N*-type calcium-channels, inhibiting the activity of a subset of neurons that includes pain-sensing primary nociceptors [20]. Despite the various attempts to develop non-opioid painkillers and anesthetics from conotoxins [21], the investigation on marine neurotoxins for biotechnological purposes is still lagging.

As marine bioprospecting expands towards other venomous marine invertebrates, attention is being given to species from temperate waters as well. It is the case of the dogwhelk *Nucella lapillus* (Gastropoda: Muricidae), an important predator of mussels and barnacles of the Western European rocky intertidal areas that is regarded as harmless and is better known for its role in the biomonitoring of tributyltins (TBS), as this anti-foulant pollutant causes imposex in females [22]. Preliminary studies suggested that this species synthesizes and secretes unidentified sulfhydryl-rich neurotoxins in salivary glands that are able to slow cardiac rates and relax the anterior byssus retractor muscle of *Mytilus* [23]. According to Ponder [24], the possession of acinus and tubular salivary glands is a plesiomorphic characteristic of neogastropods. Indeed, *Conus* and *Nucella* are anatomically similar, as they both have glandular tissue surrounded by muscle, which seemingly helps the expulsion of secretory vesicles [23,25], despite distinct feeding mechanisms, as *Nucella* feeds on bivalves by drilling shells or inserting the siphon between valves, whereas *Conus* bears a distinctive harpoon-like structure to hunt larger and more active prey, like fish. Additionally, histochemical studies confirmed the presence of cysteine-rich proteins in both species [26]. It must be noted that cysteine-rich proteins and peptides, among which CRISPs form a particular subset, are ubiquitous among animal taxa, tissue, and organs; play an important role in metabolic and physiological pathways, from redox and metal ion balance to anti-microbials; and have been persistently investigated in biomedical sciences [27]. We thus hypothesized that different cysteine-rich proteinaceous substances are secreted in different organs of dogwhelk, which implies distinct properties of interest for potential biotechnological applications. The present work is a first-level assessment that aimed at localizing the secretion of cysteine-rich proteins and peptides in dogwhelk through a comparison between multiple glandular organs, relate secretion with organs and tissue function, and compare the toxicity of secretions between organs as a frontline measure of bioreactivity.

## 2. Results

### 2.1. Inter-Organ Concentrations of Thiols

The different organs (foot, siphon, peribuccal gland mass, and digestive gland) yielded distinct concentrations of thiols normalized to total protein content (Kruskal–Wallis test: *p* = 0.000166 and *H* = 20.047; df = 3), as shown in Figure 1. The foot, here considered as reference, yielded the lowest concentration (15.5 nmol/mg), whereas the digestive gland presented the highest (115 nmol/mg). In their turn, the siphon and peribuccal glandular mass presented intermediate values (39.7 and 71.6 nmol/mg, respectively).

### 2.2. Histological Analysis of Nucella lapillus Glandular Systems

The four organs presented distinct microanatomical features and histochemical signatures (Figure 2). The foot (Figure 2a) was found to be essentially composed by connective tissue and multi-directional muscle fibers. This organ was covered by a simple columnar epithelium with elongated epithelial cells, mostly mucous cells, that secreted Alcian blue-positive substances, which is indicative of acidic mucins. In *N. lapillus*, the foot is heavily muscled with an irregular surface, as expected from an organ involved in both locomotion and adhesion to the rocky substrate. The siphon (Figure 2b) was composed of two muscle layers (circular and longitudinal), forming a canal-like structure whose lumen is externally lined by a columnar secretory epithelium. This structure also held a duct formed by a thick-walled fibrous structure lined by a single layer of epithelial cells bearing microvilli. Stem-like (replacement) cells were common in this epithelium. The peribuccal glandular mass (Figure 2c) comprises the region between the foot and digestive gland and the lodged tubular and acinar salivary glands. The latter showed an acinar structure where the acini were chiefly comprised of single-layer secretory epithelium with two types of secretory cells: mucocytes (Alcian blue-positive) and protein-secreting cells (heavily stained brownish by Weigert’s iron hematoxylin). The digestive gland (Figure 2d) presented the typical tubule-like structure of the molluscan digestive gland. The tubules (diverticula) were enclosed by loose connective tissue and were formed by a monolayered epithelium bound to a thin basal membrane. The digestive cells held numerous digestive vesicles stained by a blend of picric acid (yellow) and Weigert’s hematoxylin (Figure 2d inset), whereas the PAS-positive reaction (pink) was restricted to small vesicles.

### 2.3. Toxicological Effects of Extracts

#### 2.3.1. DNA Damage

The percentage of DNA in tail significantly varied between the gills exposed to either dilution of the crude protein extracts D1 and D2, corresponding to 100% and 50%, respectively (glm ANOVA, *F* = 2.96; *p* < 0.05). Exposure to either dilution of the digestive gland extract caused a significant increase in the percentage of DNA breaks in mussel gills (Figure 3) when compared to exposure to extracts from the reference organ, the foot (Student’s *t*-test; *p* < 0.05). Gills exposed to the most concentrated extract (D1) from the peribuccal glandular mass attained a maximum mean of 53% DNA in tail (Figure 3), followed by averages of 38% and 29% DNA in tail recorded in gills treated with digestive gland extract and siphon, respectively. Exposure to the diluted extract (D2) corresponding to a 50% dilution showed the highest value of DNA damage for gills treated with the digestive gland extract (approximately 50% DNA in tail), followed by 40% and 27% of DNA damage attained in gills exposed to siphon and peribuccal glandular mass extracts, respectively (Figure 3).

#### 2.3.2. Histological Analysis of Mussel Gills

The histopathological analysis of mussel gills revealed few differences between exposure to the different extracts and the respective controls, regardless of dilution. Generally, the exposed gills presented the expected architecture of mussel ctenidia, with a clear definition of the frontal and abfrontal zones of filaments and intact cilia (Figure 4). Only a few hemocytes were noted in hemolymphatic sinuses and adjacent tissue, indicating no significant increase in the immune response triggered by the presence of toxins or other enzymes from *N. lapillus* extracts (Figure 4a,b). Still, scarce oedemas (Figure 4b inset) were recorded in gills subjected to the treatment with the peribuccal glandular mass extract (highest concentration). Compared to the control, there was a moderate increase in the presence of apoptotic epithelial cells in gills treated with the most concentrated extracts from the peribuccal glandular mass and digestive gland (Figure 4a,b). Apoptotic cells were conspicuously identified morphologically by their highly basophilic and homogenously dense cytoplasm (yielding a high affinity to hematoxylin), as well as the characteristic blebbing of the plasmalemma.

## 3. Discussion

The current work disclosed the presence of differentially-bioactive thiol-rich proteins in various organs of the muricid marine gastropod *Nucella lapillus*. The highest concentrations of thiolic proteinaceous substances were found in peribuccal glandular mass, where the salivary glands are located, and in the digestive gland. Conversely, the foot and the siphon were the organs that showed lower quantities of thiols. The two latter organs are essentially muscular, with few glandular structures apart from mucocytes (especially in the foot). Indeed, muricids have a tubular siphonal canal whose main function is to conduct water from the pallial cavity [28]. The low concentration of thiols found in *N. lapillus* siphon likely indicates that this organ might be used only during predation process for injecting the venom rather than being primarily responsible for toxin secretion. Accordingly, our findings also indicated that the siphon has a connection with the peribuccal glandular mass, where the salivary glands are located and which should be responsible for secreting enzyme- and toxin-bearing saliva. However, it must be emphasized that in the current stage, the cystine-rich proteins in the extracts remain to be isolated and then molecularly and functionally characterized.

*Nucella lapillus* crude protein extracts hold important contents of cysteine-rich substances, highlighting the presence of sulfhydryl digestive vesicles rich in enzymes and hydrolytic groups in the digestive gland, as well as thiol-rich proteins and peptides in the salivary glands. The two organs nonetheless have distinct functions, and we therefore may expect distinct signatures of proteins and peptides. Though extracts from the digestive gland are expected to contain higher proportion of digestive enzymes, CRISPs tend to be a hallmark of animal venom glands, such as those of snakes and scorpions, and the salivary glands of mollusks. These include antigen 5 (Ag5) and pathogenesis-related 1 (PR-1) proteins, which belong to the CAP protein superfamily [2] and occur in a wide range of venomous organisms. Very few of these proteins have been functionally characterized, and there is still a lack of consistent nomenclature for CRISPs [11]. In the marine environment, besides the case of venomous cephalopods [29], recent studies have revealed the secretion of CRISP-like proteins in toxin-secreting Polychaeta, with molecular similarities to cone snail thiol-rich toxins [7,13,14], which further highlights the ubiquity of thiolic proteins. It must be noted that West [30] already suggested the venomous function of *N. lapillus* salivary glands, highlighting the presence and secretion of a compound hitherto identified as a serotonin, which, in combination with unknown bioactives, is able to cause cardiac arrest in mussels. This neurotransmitter, considered to be a relaxant toxin, is also found in sea anemone tentacles [31] and has been reported to exist in the venom of *Conus imperialis*, albeit at much lower concentrations than in *N. lapillus* [30,32].

The existence of an acinous salivary gland is compatible with a venomous function of the peribuccal glandular mass in *Nucella lapillus*. As expected, acinar cells in venomous glands are likely responsible for the synthesis and secretion of proteinaceous materials, while mucus-secreting cells provide a vehicle for toxin-delivery. According to Andrews [23] and West [25], higher concentrations of thiolic proteins in the peribuccal glandular mass of gastropods is due to the secretion of venoms rich in glycoproteins with several disulfide bonds that are also accountable for a significant portion of the crude secretion’s bioreactivity. The tubular salivary glands in *N. lapillus* are anatomically similar to cone snail venom glands [30], which are characterized by the presence of thick tubular walls composed by three different layers: circular muscle, a central lumen surrounded by columnar epithelium, and a layer of clustered flask-shaped gland cells. The subepithelial glands are specialized to secrete substances, such as glycoproteins rich in cysteine. In contrast, the acinous salivary glands are enveloped by a thin layer of connective tissue with few muscular fibers, with expulsion being made by ciliary activity via a system of narrow and ramified ducts rather than by muscle contraction. This arrangement seems to be well-conserved among the Gastropoda, at least predatory species [33], which may also indicate that the secretion of bioactives, toxins included, could be a more common feature in these organisms that anticipated. Still, at this stage, the specific function of *N. lapillus*’ various salivary glands cannot be fully ascertained due to their minute size that has hindered individualization. Further research on this aspect is needed. It should be highlighted that Muricidae is considered to be the second most significant group of venomous predatory mollusks after Conidae [34]. The most substantial difference between the two is that Conidae developed a highly specialized apparatus for delivering venom [35] that is constituted by a long and distensible proboscis and a tooth that acts as both harpoon and hypodermic needle [16]. *Conus textile*, for instance, injects venom into prey via a disposable, harpoon-like modified radular tooth [36]. In turn, members of the family Muricidae surround their prey with the foot, delivering their venomous secretions through radular action [34] or, potentially, using the siphon, which enables better stealth during hunting [30]. 

The detection of DNA damage upon exposure to extracts yielded an initial but efficient comparison of toxicity/bioreactivity. Compared to other organs, especially the foot and digestive gland (which contains digestive enzymes and is therefore expected to be able to cause significant cell and molecular damage), extracts from the peribuccal gland mass elicited the most substantial results. Together with the histopathological observation of exposed tissue, which indicated the potential promotion of cell death, the findings indicated that the salivary glands secrete cytotoxic agents, which are likely associated with important concentrations of thiolic proteins. It must be noted, however, that further dilutions are needed to establish clear dose-responsiveness. Additionally, our study focused on the cellular and molecular damage elicited by extracts, which means that physiological and metabolic imbalance resulting from exposure must still be ascertained. Even though it is not possible to identify specific substances and the specific role of thiol-rich proteins in the toxicity of crude (unfractionated) extracts at the current stage, the findings suggest important properties of the toxins secreted by the peribuccal glandular mass, among which CRISP-like proteins and peptides can be found. On the other hand, the molluscan digestive gland is a polyvalent organ that accumulates digestive functions with storage and detoxification because it is analogous of the vertebrate liver [37] and therefore may hold digestive enzymes, thionine, and anti-oxidative scavengers such as glutathione that, when combined, are responsible to the relatively high contents of thiols despite lower toxicity. Finally, the siphon appears as an extension of the mantle that conveys toxins for delivery.

In conclusion, the primary toxicological effects caused by extracts from the peribuccal extract confirm the approximate location of the *Nucella lapillus* accessory salivary glands and indicate the role of the siphon in toxin delivery but not secretion. The mixture of compounds produced by the salivary glands likely contains CRISP-like proteins and toxins, as in many animal poisons and venoms, which renders this species of temperate waters an important target for the bioprospecting of novel marine bioactives for biotechnological purposes, a focus so far almost exclusively given to tropical species.

## 4. Materials and Methods

### 4.1. Animal Collection

*Nucella lapillus* specimens (approximately 2 cm length) were collected by hand during the low tide from rocky intertidal areas in Costa da Caparica, Portugal (38°38′44.9″ N; 9°14′37.0″ W). Dogwhelks were immediately transported to the laboratory and reared in a closed-circulation aquaria mesocosm fitted with constant aeration, water filtration, and water recirculation. The aquaria were prepared with natural rocks, barnacles, and mussels collected from the same area to provide shelter and feed to the specimens. The water temperature, salinity, and photoperiod were controlled at about 18 °C, 30, and a 16:8 h light–dark cycle, respectively. Animals were acclimatized for 7–14 days until the analysis. Mussels, *Mytilus* sp., from the same area were also collected as models for the ex vivo assays.

### 4.2. Preparation of Protein Extracts

Eighteen *N. lapillus* (weight = 2.3 ± 0.6 g; length = 18.8 ± 2 mm; width = 9.7 ± 0.8 mm) were micro-dissected to collect the siphon, digestive gland, peribuccal glandular mass (which includes the salivary glands), and foot (which was taken as the reference (control), because it is essentially muscular and not glandular). Organs were separately homogenized with a pestle in cold Dulbecco’s phosphate-buffered saline (PBS) at pH 7.5 and centrifuged for 5 min, 10,000× *g*, at 4 °C. Supernatants (clarified homogenates) were pooled to obtain sufficient volumes for analysis. Each pooled sample corresponded to extracts from approximately four animals. In total, nine independent pools per organ were obtained, and these were stored at −80 °C until subsequent analyses. The total protein in pooled extracts was determined using a Nanodrop 2000 spectrophotometer (Thermo Fisher Scientific, Waltham, MA, USA).

### 4.3. Quantification of Protein Thiols

The Protein Thiol Fluorescent Detection kit (Invitrogen, Carlsbad, CA, USA) was used for the quantification of thiols, following manufacturer instructions. Briefly, in the presence of free thiols, the contact between a biological sample and the detection reagent produces a fluorescent dye that may be a product of the reaction of the cysteine content of a protein. This fluorescence product emits at 510 nm following excitation at 390 nm. Samples were diluted to 1:30 with an assay buffer, and they were incubated for 30 min with the detection reagent. Then, fluorescence was measured using an Infinite 200 fluorescence microtiter plate reader (Tecan, Männedorf, Switzerland) using a filter pair for excitation at 485 nm and emission at 535 nm. Results are expressed as nmol cysteine equivalent/mg total protein. 

### 4.4. Histology and Histochemistry

To maximize the preservation of the internal structure, whole *Nucella lapillus* specimens were infiltrated by injection of Davidson’s fixative (9–10% *v*/*v* formalin, 10% *v/v* glacial acetic acid, and 30% ethanol prepared in MilliQ-grade ultrapure water > 16 MΩ cm) through a hole in the apex, spiral zone, and beneath the operculum. Samples were then immersed in a fixative, de-shelled after about 15 min, and again infiltrated in a fixative for 24 h at room temperature. Afterwards, the *N. lapillus* organs were washed in distilled water (4 × 15 min) and then dehydrated through a progressive series of (aqueous) ethanol: 70% (1 × 30 min), 95% (2 × 15 min), and 100% (3 × 30 min). Intermediate infiltration and embedding were done with xylenes (3 × 15 min) and molten paraffin (overnight), respectively. All samples were sectioned at a 5 µm thickness using a RM 2125 RTS rotary microtome (Leica Microsystems, Wetzlar, Germany).

Sections were stained using a tetrachrome technique to enhance structural detail. This method combines Alcian blue for acidic sugars and mucins, Periodic Acid - Schiff’s (PAS) for neutral sugars plus Weigert’s iron hematoxylin and picric acid as counter-stains for muscle fibers and cytoplasm, respectively [38]. All slides were dehydrated, cleared with xylenes, and mounted with DPX resin. Observations were performed using a DM 2500 model microscope (Leica Microsystems, Wetzlar, Germany).

### 4.5. Assays with Mytilus Gills

#### 4.5.1. Experimental Design

Ex vivo bioassays using *Mytilus* gills were performed to investigate DNA and tissue damage resulting from exposure to crude extracts from the different *N. lapillus* organs. For the purpose, valves of live mussels (approximately 40 mm in length) were carefully separated to retain the integrity of gills, and the visceral mass was swiftly removed. Assays began immediately after excision. One valve was used as a test sample (*N. lapillus* extracts diluted in PBS, as described above), and the other was used as its respective control (PBS only). For all crude extracts, the test valve was exposed to 1 mL of D1 or D2 dilutions, corresponding to 1 mg/mL and 0.5 mg/mL total protein, respectively, while the control was treated with 1 mL of PBS [39]. Test valves and controls were assayed for 1 h and processed immediately afterwards for subsequent histopathological analysis (as described above) and comet assay. All assays were made in triplicate (n = 3).

#### 4.5.2. Comet Assay

Damage to DNA was evaluated using an adaptation of the alkaline single-cell gel electrophoresis (comet) assay developed by Singh et al. [40] and adapted by Raimundo et al. [41] and Martins and Costa [42]. In brief, freshly-harvested gill samples were minced with pliers in 700 mL of cold PBS and centrifuged for 1 min at 1500 rpm. The supernatant (clear cell suspension) was embedded in 1% m/v molten (37–40 °C) low-melting-point agarose (LMPA) prepared in PBS. Afterwards, two 80 µL drops of LMPA cell suspensions were placed on slides pre-coated with 1.2% m/v of normal-melting-point agarose (dried for at least 48 h) and covered with a coverslip. After LMPA solidification (15 min at 4 °C), coverslips were removed and the slides were immersed in a cold lysis buffer (0.45 M NaCl, 40 mM EDTA, and 5 mM Tris pH 10) for 1 h. Slides were then placed in cold electrophoresis buffer (0.2 mM EDTA and 0.3 M NaOH at pH 12) for 40 min to allow for DNA-unwinding and the expression of alkali-labile sites. Electrophoresis was run at 25 V for 30 min at 4 °C. Slides were neutralized in 0.2 M Tris-HCl (pH 7.5) and dried with methanol for archiving before analysis. Rehydrated slides were stained with GreenSafe (Nzytech, Lisbon, Portugal) [42] and analyzed with a DM 2500 LED microscope adapted for epifluorescence with an EL 6000-light source (Leica Microsystems). Scoring was done with CometScore 1.6 (TriTek, Sumerduck, VA, USA), with 100 nucleoids being analyzed per slide. The percentage of DNA in tail was considered as a direct measure of DNA damage [36].

#### 4.5.3. Histopathology

The histopathological procedure applied to the gills of *Mytilus* sp. followed the above-mentioned process, i.e., after the bioassay, gills were collected and preserved in Davidson’s fixative for 24 h. Afterwards, samples were washed in distilled water, dehydrated, and embedded with molten paraffin, as stated above. The 5-µm-thickness sections of mussel gills were stained with the hematoxylin (Harris’) and eosin (H&E) staining protocol, as detailed in [43].

### 4.6. Statistical Analysis

The normality and homoscedasticity of data were assessed by the Shapiro–Wilk and the Levene’s test, respectively. Non-parametric procedures were based on the Kruskal–Wallis H (ANOVA-by-ranks) test, followed by the Dunn’s test, for post-hoc comparisons. When the assumptions for parametric analyses were met, an F test-based ANOVA was conducted, followed by the post-hoc Tukey’s honestly significant difference (HSD) test. Complementary comparisons were done with Student’s *t*-test. All analysis were done using the software R 3.6.1 [44]. A significance threshold of 0.05 was considered for all analysis.

## Figures and Tables

**Figure 1 marinedrugs-19-00276-f001:**
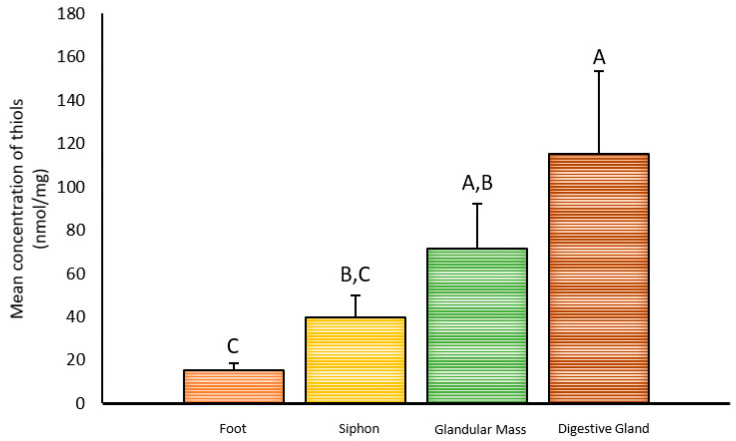
Concentration and distribution of thiols in *N. lapillus*. The results are expressed as nmol cysteine equivalents per mg total protein (mean + standard deviation) in different organs: foot (reference organ), siphon, (peribuccal) glandular mass, and digestive gland. Different letters indicate significant differences (Dunn’s test; *p* < 0.05).

**Figure 2 marinedrugs-19-00276-f002:**
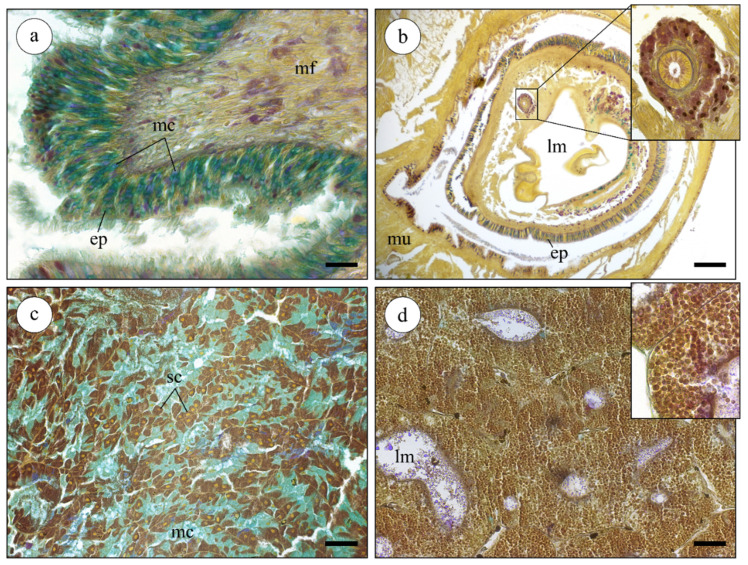
Histological sections (paraffin) of multiple organs of the dogwhelk *N. lapillus* stained with a tetrachrome stain. (**a**) Cross section of the foot, showing the multi-directional muscle fibers (mf) and the epithelium (ep) where the mucous-secreting cells (mc) are stained greenish-blue. (**b**) Siphon section localized close to the digestive gland. Note the existence of an epithelium (ep), adjacent muscle fibers (mu), and the lumen (lm). Inset: detail of the duct with a single layer of epithelial cells. (**c**) Section across a salivary gland of the peribuccal glandular mass, showing the two types of secretory cells: protein-secreting cells (sc) with brownish cytoplasm (from Weigert’s hematoxylin) due to high number of ribosomes and mucous-secreting cells (mc), which are Alcian blue-positive (indicating acidic mucins). (**d**) Typical tubular-structure of the digestive gland. The digestive gland ducts are formed by a large number of small digestive vesicles (inset). Scale bars: (**a**,**c**,**d**) 50 µm; (**b**) 200 µm.

**Figure 3 marinedrugs-19-00276-f003:**
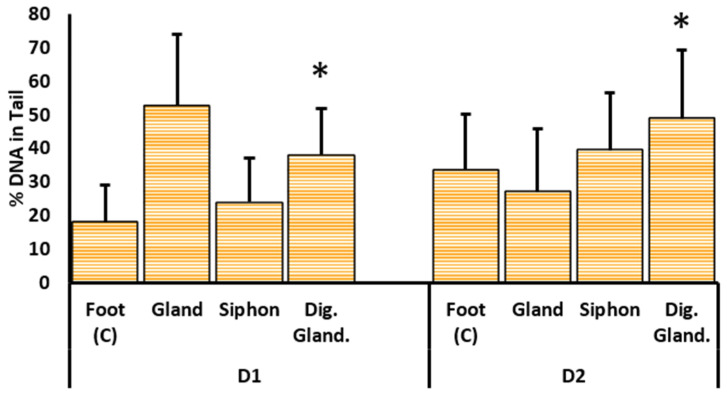
Comet assay results expressed as %DNA in tail from the gills of mussels exposed ex vivo to crude protein extracts of two dilutions from different organs—foot, peribuccal glandular mass (gland), siphon, and digestive gland of *N. lapillus*. Results are expressed as means + standard deviation, and they correspond to the nominal dilutions of 100% (D1) and 50% (D2). * indicates significant differences to the reference organ, the foot (Student’s *t*-test *p* < 0.05).

**Figure 4 marinedrugs-19-00276-f004:**
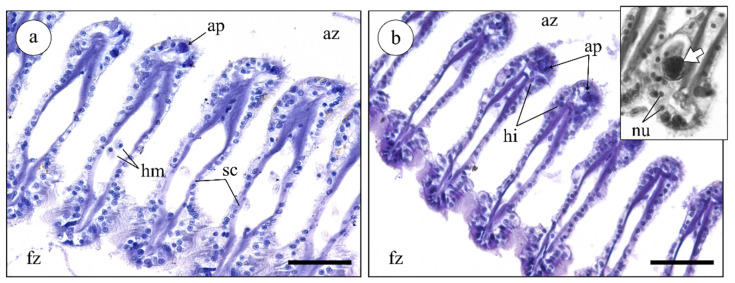
Representative histological gill sections of *Mytilus* sp. treated with the most concentrated (D1) extract from the *N. lapillus* peribuccal glandular mass (H&E). (**a**) Transversal cut of a control gill (treated with PBS only). Fz: frontal zone; az: abfrontal zone; sc: supporting cartilage. (**b**) Transversal cut of gill exposed to extract. The presence of hemocytes (hm), hemolytic infiltration (hi), and apoptotic cells (ap) was verified in both sections, but the frequency was higher in exposed gills. Inset: detail of an edematous (fluid-retaining) area with apoptotic cells (white arrow) and normal nuclei (nu) observed on the same section. Scale bars: 50 µm.

## Data Availability

Data is contained within the article.

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
