# Peer review of "Localization and Bioreactivity of Cysteine-Rich Secretions in the Marine Gastropod Nucella lapillus"

_marinedrugs, 2021, doi:10.3390/md19050276_

Round 1

Reviewer 1 Report

This is a short and sweet paper, enjoyable at reading, well presented, with data supporting the conclusions, which provides also an ecological insight. I only recommend the authors to revise carefully the text due to some typos or inacurracies, and add the paper below as they mention cnidarians in particular.

L36: As the authors specifically mention cnidarians, I suggest the following review would be very appropriate:

D'Ambra, I. and C. Lauritano (2020). "A Review of Toxins from Cnidaria." Marine Drugs 18(10): 507.

                Fig 1: I suggest to remove the little squares with colors and add the name of the body part directly under the corresponding histogram

L95: Nucula lapillus goes in italics

L146: probably “gills” and “mussel” should be inverted

L165: “the current” what? I think the subject is missing in this sentence

Reviewer 2 Report

Focusing on the usefulness of the toxins secreted by marine organisms, the authors have demonstrated the presence and activity of CRISP in crude extracts from each organ of the highly toxic mussel, Nucella lapillus. Although it is a very interesting draft, the reviewers think that improving the following points will make the treatise more understandable to the reader.

  1. Regarding Fig. 3, please explain in consideration that the activity of "Gland extract" in Comet assay is significantly different between D1 and D2 compared with the values of extracts from other tissues.
  2. Regarding the difference in apoptosis cells depending on the presence or absence of the action of the D1 extract shown in FIG. 4, it is difficult to understand the difference in this figure. I want you to make the figure easier to understand, such as specific staining of apoptotic cells.

3. In the discussion section, please indicate the limitation of the experimental data used in this paper
